# One View Per City for Buildings Segmentation in Remote-Sensing Images via Fully Convolutional Networks: A Proof-of-Concept Study

**DOI:** 10.3390/s20010141

**Published:** 2019-12-24

**Authors:** Jianguang Li, Wen Li, Cong Jin, Lijuan Yang, Hui He

**Affiliations:** 1College of Information and Communication Engineering, Communication University of China, Beijing 100024, China; lijianguang@cuc.edu.cn (J.L.); jincong0623@cuc.edu.cn (C.J.); 2State Key Laboratory of Media Convergence and Communication, Communication University of China, Beijing 100024, China; 3Shenzhen College of Advanced Technology, University of Chinese Academy of Sciences, Shenzhen 518055, China; wen.li@siat.ac.cn; 4Shenzhen Institutes of Advanced Technology, Chinese Academy of Sciences, Shenzhen, Guangdong 518055, China; 5Ocean College, Minjiang University, Fuzhou 350108, China; 2611@mju.edu.cn; 6College of Information Technology, Beijing Normal University, Zhuhai 519087, China; 7Engineering Research Center of Intelligent Technology and Educational Application, Ministry of Education, Beijing 100875, China

**Keywords:** remote-sensing, one view per city, buildings segmentation, fully convolutional network

## Abstract

The segmentation of buildings in remote-sensing (RS) images plays an important role in monitoring landscape changes. Quantification of these changes can be used to balance economic and environmental benefits and most importantly, to support the sustainable urban development. Deep learning has been upgrading the techniques for RS image analysis. However, it requires a large-scale data set for hyper-parameter optimization. To address this issue, the concept of “one view per city” is proposed and it explores the use of one RS image for parameter settings with the purpose of handling the rest images of the same city by the trained model. The proposal of this concept comes from the observation that buildings of a same city in single-source RS images demonstrate similar intensity distributions. To verify the feasibility, a proof-of-concept study is conducted and five fully convolutional networks are evaluated on five cities in the Inria Aerial Image Labeling database. Experimental results suggest that the concept can be explored to decrease the number of images for model training and it enables us to achieve competitive performance in buildings segmentation with decreased time consumption. Based on model optimization and universal image representation, it is full of potential to improve the segmentation performance, to enhance the generalization capacity, and to extend the application of the concept in RS image analysis.

## 1. Introduction

Buildings segmentation using remote-sensing (RS) images plays an important role in monitoring and modeling the process of urban landscape changes. Quantification of these changes can deliver useful products to individual users and public administrations and most importantly, it can be used to support the sustainable urban development and to balance both the economic and environmental benefits [1,2,3,4,5]. Via analyzing the data source of Landsat TM/ETM+ in 1990s, 2000s and 2010s, a study estimated China’s urban expansion from the urban built-up area, the area of croplands converted into urban as well as the speed of urbanization in different cities [6]. It gave clues on the relationship among urbanization, land use efficiency of urban expansion and population growth. Moreover, these factors are highly related to carbon emissions, climate change and urban environmental development that facilitate urban planning and management [7].

A large number of computational methods have been developed for RS image segmentation [8], since manual annotation of high-resolution RS images is not only time-consuming and labor-intensive but also error-prone, and therefore, to develop high-performance methods is urgent for RS image analysis. Techniques for image segmentation can be broadly grouped into semi- and full-automatic methods. Semi-automatic methods require user assistance and graph cuts is one of the most notable methods [9,10,11,12]. The method takes intensity, textures and edges of an image into consideration and after some pixels are manually localized in background, foreground or unknown regions, it addresses the problem of binary segmentation by using Gaussian mixture models [13]. Finally, one shot of object segmentation is achieved from iterative energy minimization. Wang et al. [14] integrated a graph cuts model into spectral-spatial classification of hyper-spectral images and in each smoothed probability map, the model extracted the object to a certain information class. Peng et al. [15] took advantage of a visual attention model and a graph cuts model to extract the rare-earth ore mining area information using high-resolution RS images. Notably, semi-automatic methods enable a user to incorporate prior knowledge, to validate results and to correct errors in the process of iterative image segmentation.

It is imperative to develop full-automated methods for RS image analysis, particularly when the spatial and temporal resolution of RS imaging has been considerably and continuously increased. The approaches for full-automated segmentation of RS images can be divided into conventional methods and deep learning (DL) methods in general. The former is developed based on the analysis of pixels, edges, textures and regions [8,16,17]. Hu et al. [18] designed an approach that consisted of algorithms for determination of region-growing criteria, edge-guided image object detection and assessment of edges. The approach detected image edges with embedded confidence and the edges were stored in an R-tree structure. After that, initial objects were coarsely segmented and then organized in a region adjacency graph. In the end, multi-scale segmentation was incorporated and the curve of edge completeness was analyzed. Interestingly, some methods incorporate machine learning principles and recast RS image segmentation as a pixel- or region-level classification problem [19,20]. However, parameters in most approaches are set empirically or adjusted toward high performance and thus, the generalization capacities might be restricted.

Recently, DL has revolutionized image representation [21], visual understanding [22], numerical regression [23] and cancer diagnosis [24]. Many novel methods have been developed for RS image segmentation [25,26,27,28,29]. Volpi and Tuia [30] presented a fully convolutional neural network (FCN) and it achieved high geometric accuracy of land-cover prediction. Kampffmeyer et al. [31] incorporated median frequency balance and uncertainty estimation which aimed to address class imbalance in semantic segmentation of small objects in urban images. Langkvist et al. [32] compared various design choices of a deep network for land use classification and the land areas were labeled with vegetation, ground, roads, buildings and water. Wu et al. [33] explored an ensemble of convolutional networks for better generalization and less over-fitting and furthermore, an alignment framework was designed to balance the similarity and variety in multi-label land-cover segmentation. Alshehhi et al. [34] proposed a convolutional network model for the extraction of roads and buildings and to improve the segmentation performance, low-level features of roads and buildings were integrated with deep features for post-processing. Vakalopoulou et al. [35] used deep features to represent image patches and support vector machine was employed to differentiate buildings from the background regions. Gao et al. [36] designed a deep residual network and it consisted of a residual connected unit and a dilated perception unit and in the post-processing stage, a morphologic operation and a tensor voting algorithm were employed. Yuan [37] proposed a deep network with a final stage that integrated activations from multiple preceding stages for pixel-wise prediction. The network introduced the signed distance function of building boundaries as the output representation and the segmentation performance was improved.

It has achieved promising results by using DL methods for automated objects segmentation in RS images. However, DL requires considerable data for hyper-parameter optimization [38]. In the field of RS imaging, to collect sufficient images with accurately annotated labels is challenging, since a lot of objects of interest are buried in a complex background and a large area mapping [39]. To address this issue, a concept of “one view per city” (OVPC) is proposed. It explores to make the most of one RS image for parameter settings in the stage of model training, with the hope of handling the rest images of the same city by the trained model. As such, challenges could be relieved to some extent. It requires only one image per city and thus, time and labor can be reduced in the labeling of ground truth for specific purposes as well as the model training. Moreover, an algorithm is trained and tested on images from the same city and thus, intrinsic similarity between the foreground and background regions could be well explored. In fact, the concept comes from the observation that buildings of a same city in singe-source RS images illustrate similar intensity distributions. To verify its feasibility, a prove-of-concept study is conducted and five FCN models are evaluated in the segmentation of buildings in RS images. In addition, five cities in the Inria Aerial Image Labeling (IAIL) database [40] are analyzed.

The rest of this paper is organized as follows. Section 2 shows the observation that the buildings of a same city acquired by a same sensor show similar intensity distributions. Section 3 describes the involved FCN models and then introduces the data collection, experiment design, performance metrics and algorithm implementation. Section 4 demonstrates experimental results and Section 5 discusses the findings. This proof-of-concept study is concluded in Section 6.

## 2. One View Per City

The proposal of the concept “one view per city” (OVPC) comes from the observation that most of buildings from a same city acquired by a same sensor demonstrate a similar appearance in RS images. To show the observation, the IAIL database [40] is analyzed. Specially, the appearance of buildings is quantified with the distribution of pixel intensities in the annotated regions in RS images. As shown in Figure 1, each row stands for a city (Austin, Chicago, Kitsap County, Western Tyrol and Vienna), each column indicates the red, green or blue channel of images, and each plot shows the intensity distributions of all 36 images. Moreover, in each plot, the horizontal axis shows the intensity range ([1, 255]), and the vertical axis shows the number of pixels to each intensity value.

Pair-wise linear correlation coefficients (LCCs) of distributions of pixel intensities are calculated. In Figure 2, each row stands for a city, each column indicates the red, green and blue channel of images and each plot shows a LCC matrix. Note that in each plot, both the horizontal and the vertical axis shows the image index. It is observed that when 0.5 is defined as the threshold of LCC values, 98.61% RS image pairs from the city Austin shows a higher correlation, followed by Vienna (97.69%), Western Tyrol (82.72%), Kitsap County (75.15%) and Chicago (57.87%).

The observation can be visually perceived. Figure 3 shows RS images of the 6th and 21st (**A**ustin, noted as A06 and A21), the 14th and 26th (**C**hicago, noted as C14 and C26), the 33rd and 36th (**K**itsap County, noted as K33 and K36), the 13th and 22nd (Western **T**yrol, noted as T13 and T22), and the 5th and 33rd (**V**ienna, noted as V05 and V33). It is found that these cities can be visually distinguished from each other by comparing the major appearances of buildings.

Both the quantitative comparison (Figure 2) and the visual observation (Figure 3) suggest that the buildings of a same city illustrate a similar appearance in single-source RS images. Intuitively, this kind of information redundancy can be utilized to address the issue of limited data in DL based RS image analysis. Therefore, the proposal of OVPC might benefit pixel-wise segmentation of buildings in RS images.

## 3. A Proof-of-Concept Study

To verify the feasibility, a proof-of-concept study is conducted. The task is to segment buildings in RS images using OVPC based FCN models. In short, five FCN models are evaluated on RS images of five cities in the IAIL database [40].

### 3.1. Fully Convolutional Neural Network

In general, FCN architectures consist of an encoder and a decoder network symmetrically and Figure 4 illustrates a classic network, U-Net [41]. The main blocks are convolutional layers, pooling layers, up-convolutional layers and in particular, the concatenate parts propagate information from the encoder to the decoder which keeps the visual information fidelity during image restoration.

The models FCN8 and FCN32 [42] modify pre-trained neural networks for pixel-wise prediction. In particular, a skip architecture is added which embodies not only deep, coarse, sematic information, but also shallow, fine, appearance messages. It is the prototype of encoder-decoder architectures for pixel-wise image segmentation. In addition, both FCN8 and FCN32 are based on the 16-layer VGG net [43] and the difference comes from the restoration position of the skip architecture.

The model SegNet [44] is made up of an encoder network, a decoder network and a pixel-wise classification layer. Topologically, the encoder network is identical to the 13-layered VGG net [43]. Its decoder makes use of pooling indices in the max-pooling step of the corresponding encoder to perform non-linear up-sampling. Because up-sampled maps are sparse, the SegNet further employs trainable convolutional filters and produces dense feature maps for pixel-wise labeling.

The model TernausNet [45] is also an encoder-decoder architecture. It employs the VGG network [43] and contains 11 sequential layers as its encoder. Comparing three weight initialization schemes, experimental results suggest that the VGG network pre-trained on ImageNet [46] achieves relatively better performance in image segmentation.

The U-Net [41] takes advantage of a contracting path for context capturing and a symmetric expanding path for precise localization. It allows for the propagation of context information to higher resolution layers by using these previously extracted feature channels. In particular, a weighted loss function is additionally used to separate the background regions between touching objects. It has been widely used in RS image analysis, such as buildings segmentation [47], damage mapping [48] and crop mapping [49].

Among the five encoder-decoder architectures, four (FCN8, FCN32, SegNet and TernausNet) utilize the VGG net [43] with different number of successive layers. The 16-layer VGG net is shown in Figure 5 and it consists of 13 convolutional layers, 5 pooling layers, 3 full-connection layers and 1 softmax layer.

### 3.2. Data Collection

The IAIL database [40] is analyzed. It contains a total of 360 RS images with regard to 10 cities and each city is with 36 images (https://project.inria.fr/aerialimagelabeling/). Images are formatted with GeoTIFF, the spatial resolution of aerial orthorectified color images is 0.3 m and the image matrix size is [5000, 5000]. Furthermore, manual labels of five cities are provided and images are annotated into building and not
building regions. Therefore, in this study, the images with ground truth are used. The involved satellite images cover 5 cities and 405 km${}^{2}$. These cities are Austin (America), Chicago (America), Kitsap County (America), Western Tyrol (Austria) and Vienna (Austria). The RS images demonstrate different density of urban settlements, various urban landscapes and illumination [50]. For instance, buildings in Kitsap County are sparsely scattered, while buildings in Chicago are densely distributed (Figure 3).

The area of buildings is further concerned. To make it clear, a parameter is defined which aims to describe the coverage of buildings regions in each image. The parameter, buildings region ratio (brr), is computed as the pixel number of buildings (*B*) over that of each image (*I*) as shown in Equation (Equation 1),
(1)$$brr=\frac{\left||B|\right|}{\left||I|\right|},$$
where ||·|| denotes the number of pixels.

Figure 6 shows the distributions of brr values where the x-axis indicates cities and the y-axis is brr values. In violin plots, solid lines denote median values and dashed lines stand for the 1st and the 3rd quartiles. The mean of brr values is 0.1504 ± 0.0477 (Austin), 0.2433 ± 0.0599 (Chicago), 0.0485 ± 0.0420 (Kitsap County), 0.0584 ± 0.0378 (Western Tyrol) and 0.2884 ± 0.1153 (Vienna). It indicates that the city Vienna has the highest buildings density, followed by Chicago and Austin, while RS images of Western Tyrol and Kitsap County are with buildings thinly scattered.

### 3.3. Experiment Design

Based on the concept of OVPC, one image is used for parameter tuning and the rest images of the same city are used for testing (Table 1). Specifically, to each city, 5 images are randomly selected and each plays as the input for model training. After the model is trained, the rest 35 images are tested (intra-city test). Moreover, images from other cities (a total of 144 images) are also tested using this trained model (inter-city test).

### 3.4. Performance Metrics

Two metrics are used to evaluate the segmentation performance. One is segmentation accuracy (ACC) and it measures the percentage of correctly classified pixels (Equation (Equation 2)). The other metric is intersection over union (IoU) which is the ratio of the number of pixels labeled as buildings in both the prediction and the reference divided by the number of pixels labeled as pixel in the prediction or the reference (Equation (Equation 3)). Given the prediction result *P* and the reference *S*, the metrics of ACC and IoU are respectively defined as
(2)$$ACC=\frac{\left|P\cap S\right|}{\left|S\right|},$$
and
(3)$$IoU=\frac{\left|P\cap S\right|}{\left|P\cup S\right|},$$
where |·| denotes the number of foreground pixels in the binary images.

### 3.5. Algorithm Implementation

Algorithms were run with Linux system (Ubuntu 16.04.10) on 3 workstations. The workstations are all embedded with 16 Intel(R) Xeon(R) CPU (3.00 GHz), 64 GB DDR4 RAM and one GPU card (TITAN X (Pascal), 12 GB). FCN models are available online (https://github.com/divamgupta/image-segmentation-keras). Deep networks are implemented with Keras (https://keras.io/) (Python 2.7.6) and the backend is Tensorflow (https://www.tensorflow.org/).

In detail, images are cropped to [2${}^{12}$, 2${}^{12}$] and the boundary of the original images is discarded. A total of 15,625 patches are extracted from each cropped image. Note that 2${}^{5}$ pixels are overlapped between successive patches. To deep models, the size of input patches is defined as [2${}^{7}$, 2${}^{7}$] and the pixel intensities in each patch are linearly scaled to [0, 1]. Parameters are set as follows. The binary cross entropy is set as the loss function, Adam as the optimizer, the learning rate is 10${}^{-4}$, the batch size is 2${}^{6}$ and the number of epochs is 10${}^{2}$. Other parameters are set as default. In addition, neither fine-tuning nor data augmentation are used.

## 4. Results

### 4.1. Intra-City Test

Table 2 shows the mean and standard deviation of ACC values (mean ± std, %) and the highest mean values in each intra-city test are bold-faced. In general, it shows that major building regions are correctly predicted (>80%) except the SegNet on Chicago (79.41 ± 7.18%). Based on the performance analysis of five FCN models with regard to different cities, it is found that the buildings in Kitsap County might be the easiest to be segmented (>94%), followed by Austin (≈90%), West Tyrol and Vienna, and the last one is Chicago. Notably, Unet achieves the overall best segmentation results.

Table 3 shows the IoU values (mean ± std, %) and the highest mean values in each intra-city test are in bold. It is found that the performance of buildings segmentation is moderate. The worst result is from FC32 on West Tyrol (54.22 ± 9.75%) and the best is from Unet on Vienna (72.11 ± 6.43%). Based on the performance analysis with regard to different cities, it is observed that the buildings in Vienna and Austin might be much easier to be outlined (>64%). It is worth mentioning that Unet gets best results on three cities (Chicago, West Tyrol and Vienna) and in general, superior performance over other FCN models (>63%).

Figure 7 shows the values of performance metrics with regard to different models and cities. The rows from top to bottom denote FCN models of FCN8, FCN32, SegNet, TernausNet and U-Net, and the columns from left to right correspond to metrics of ACC and IoU, respectively. In each box and whisker plot shown with the minimum and maximum values, the x-axis indicates the cities, the y-axis denotes the metric values (%).

At present, tens of studies are conducted on the IAIL database [50,51,52,53,54,55,56,57,58]. Several studies [50,51,52,53,54] follow a common practice as [40] that the first 5 images of each city are used for testing. In other words, there are 31 images for training and the rest 5 images for testing regarding each city. Current outcomes on the IAIL database are summarized in Table 4. The highest metric values are in bold. It is observed that the generative adversarial network [59] with spatial and channel attention mechanisms (GAN-SCA) [54] achieves the best overall segmentation results.

Table 4 reports the results of OVPC based Unet. To RS images of one city, the model uses 1 image for training and the rest 35 for testing. Comparing the results of OVPC based Unet with the recent best outcome (GAN-SCA [54]), the ACC drop is between 2.62% (West Tyrol) and 4.60% (Chicago) and IoU decrease is between 5.32% (Kitsap County) and 11.41% (Austin). On the other hand, the Unet achieves competitive or superior performance over the baseline of MLP network [40]. It is found that ACC values slightly reduce from 0.96% (West Tyrol) to 6.02% (Chicago), while IoU values obviously increase for the cities of Austin (8.40%), Kitsap County (11.81%) and West Tyrol (9.68%).

Table 4 also lists the performance of Unet that takes 31 images of each city for training. Note that the architecture and parameter settings of implemented Unet models are the same. The comparison indicates that using more images of a city for training leads to slight increase (2% to 5%) on both ACC and IoU values.

### 4.2. Inter-City Test

The generalization capabilities of deep networks on IAIL database are concerned [40,60]. The results of inter-city test using OVPC based Unet are shown in Table 5 and Table 6. The ACC values indicate that the segmentation performance decreases when using the model trained on one city to predict another city. Specifically, the model trained on West Tyrol gets the minimum decrease when it tests on other cities (≤5.28%), followed by the model trained on Chicago (≤5.72), Kitsap County (≤5.80), Vienna (≤7.42) and Austin (≤8.60).

Comparing the IoU values in Table 6 demonstrates that the performance decreases when using the model trained on one city to predict another city. Specifically, the model trained on Chicago gets a relatively small decrease (≤7.82%), followed by the model trained on Kitsap County (≤10.48), Austin (≤14.87), West Tyrol (≤15.53) and Vienna (≤16.88).

### 4.3. Time Consumption

Different implementations lead to various time consumption. In this study, one epoch takes about 3.2 min to the OVPC based Unet and it costs 5.4 h to complete the whole model training. While using the 31 RS images as the input for training (Unet ${}^{a}$), the time per epoch increases to 21.0 min and the complete training takes about 35.0 h. In addition, to predict a whole RS image takes ≈6.2 min, including thresholding and small patch merging.

## 5. Discussion

This study proposes the concept of OVPC. It explores to relieve the requirement of a large-scale data set to some degree in DL based RS image analysis. The concept proposal is inspired by the observation that buildings of a same city share similar appearance in single-source RS images. This study illustrates the observation qualitatively (Figure 2) and quantitatively (Figure 3). Furthermore, the concept is verified on buildings segmentation in RS images via DL methods and five FCN models are evaluated on RS images of five cities in the IAIL database. At last, its pros and cons are analyzed through intra-city test, inter-city test and time consumption.

The building regions from a same city shares similar intensity distribution in RS images. The quantitative analysis (pair-wise LCCs) of intensity histograms indicates that building regions between the RS images from Austin correlates strongly, followed by the images from Vienna (Figure 2). The similar appearance can also be observed from visual comparison (Figure 3). Therefore, the proposal of the concept is reasonable and it is possible to use the similarity in intensity distributions in RS image analysis. And further, it might be able to reduce time and labor in data annotation, algorithm design and parameter optimization.

In this proof-of-concept study, intra-city test shows that the OVPC-based Unet achieves superior performance over other networks (Table 2 and Table 3), and its generalization capacity is inadequate as shown in inter-city test (Table 5 and Table 6). The result of this study is similar to the findings in [60] which suggests the Unet architecture is well suited for image dense labeling, while outcome of the cross-city test is not satisfactory yet (ACC ≈ 95% and IoU ≈ 73%). In particular, in this study, the intra-city test shows that FCN models achieve moderate to excellent performance. The metric ACC indicates that RS images are correctly portioned into buildings and not
buildings (>80%), in particular in the images from Kitsap County and Austin (Table 2), while the metric IoU shows that background regions are misclassified into buildings regions and several values are less than 60%, such as SegNet on Chicago (Table 3). Unsurprisingly, the inter-city test finds out that it is challenging to accurately and precisely isolate buildings regions of one city by using an OVPC based Unet model trained on another city (Table 5 and Table 6). In detail, ACC values show slight decrease (Table 5), while IoU values reveal around 10% drop in buildings segmentation (Table 6).

To OVPC-based deep models, the reason of promising results in intra-city test mainly comes from the high-performance image representation of deep networks and these networks can represent complex patterns with hierarchical features. Moreover, these models have demonstrated capacities of pixel-wise semantic segmentation in various fields, such as computer vision [46] and biomedical imaging [42,44,45]. In particular, OVPC utilizes a large number of patches (i.e., 15,625) from one image as the input of deep networks and then, the information redundancy of building appearance is further used for the segmentation of buildings in other images from the same city. On the other hand, reasons for moderate IoU values are manifold. First, OVPC decreases the capacity of image representation of DL models due to limited training samples. Furthermore, the area of buildings regions over the RS image (i.e., brr) is tiny, such as 0.0485 ± 0.0420 of Kitsap County (Figure 6), that dramatically imposes difficulties on deep networks to learn effective representation. Second, some key parameters, such as loss function, should be fine-tuned or carefully designed [60]. From the technical point of view, data augmentation, batch normalization and transfer learning can be further integrated to improve the segmentation performance. In addition, RS image segmentation is a long-standing problem. Due to the unique cultures of western countries, buildings in RS images are distributed with different sizes and shapes. For instance, the buildings in Kitsap County are scattered, while buildings in Chicago are densely distributed and most buildings are small in size [54]. Furthermore, boundaries between buildings are ambiguous that makes accurate segmentation challenging.

This study suggests that OVPC is beneficial to RS image analysis. It requires one RS image for model training and thereby, the time and labor in manual annotation can be reduced. To annotate a large scale of images, in particular high-resolution RS images, is always an expensive task, and cross checking should be carried out to minimize the risk of false annotation [61]. To address the challenge, few-shot learning becomes a hot topic in RS image classification [62,63,64]. Impressively, Song and Xu explored zero-shot learning for automatic target recognition in synthetic aperture radar (SAR) images [65]. Moreover, using one single RS image for model training might save computing resources and decrease time cost in model training. Under the context of a GPU card with 12 GB memory, the experimental design, model implementation and time cost should be fully considered when a large number of samples are as input for training. Based on the IAIL database, when using the entire data set as the input, one epoch would last more than 2.5 h which is inconceivable [50]. In this study, one epoch takes about 2 to 5 min dependant on the FCN model and subsequently, a total of 3.5 to 8.5 h to complete the whole model training. Based on the Unet with same architecture and parameter settings, the time cost is further compared when using different numbers of images as the input for model training. It finds out that one epoch takes about 3.2 min to the OVPC based Unet and ≈21.0 min to the Unet with 31 images as its input. In other words, the proposed OVPC based Unet achieves competitive performance with dramatically decreased time consumption. More importantly, the performance of OVPC based RS image analysis could be further improved when advanced networks are used, which can be observed by comparing the results in Table 2 and Table 3 with the recent outcomes in Table 4. This study uses original networks, such as U-Net [41], and advanced networks [50,51,52,53,54] can improve the segmentation results (Table 4). In detail, multi-task SegNet [52] embeds a multi-task loss that can leverage multiple output representation of the segmentation mask and meanwhile bias the network to focus more on pixels near boundaries; MSMT [53] is a multi-task multi-stage network that can handle both semantic segmentation and geolocalization using different loss functions in a unified framework; and GAN-SCA [54] integrated spatial and channel attention mechanisms into a generative adversarial network [59]. At last, the proposed concept can be further extent to different types of RS images and applications. RS image segmentation is indispensable to measure urban metrics [66,67], to monitor landscape changes [68] and to model the pattern and extent of urban sprawl [69]. It is also important to define urban typologies [5], to classify land use [4], to manage urban environment [2] and to support sustainable urban development [6,7]. While for accurate decision making, diverse techniques should be involved [70,71,72,73,74], such as LiDAR and aerial imagery.

On further improving the performance of OVPC based buildings segmentation in RS images, additional techniques could be considered. Above all, the concept requires images should be acquired from a same imaging sensor. To enhance its generalization capacity, universal image representation is indispensable which aims to transform the source and the target images into a same space. For instance, Zhang et al. [75] improved the Kalman filter and harmonized multi-source RS images for summer corn growth monitoring. Notably, generative adversarial network [59] has been used to align both panchromatic and multi-spectral images for data fusion [76]. These methods provide insights on how to generalize the proposed concept into multi-source RS image analysis. Moreover, data augmentation is helpful to enhance representation capacity of deep networks (https://github.com/aleju/imgaug). Data transformation, shape deformation and other various distortions can be used to represent buildings characteristics. Attention can also be paid to architecture design, batch normalization and parameter optimization. Besides, transfer learning is suggested to enhance network performance [77] and it requires domain adaption to balance the data distributions of the source and the target domain [78]. In addition, except for the appearance, it is potential to model buildings from shape and texture and to enrich our understanding of urban buildings in RS images. Last but not the least, post-processing strategies can be employed and prior knowledge and empirical experiences become helpful.

This study has some limitations. At first, the pros and cons of the concept OVPC are not explicitly revealed. It is better to use each of the 180 RS images (36 images per city × 5 cities) for OVOC based buildings segmentation, while that would cost more than 1000 h for model training (≈6 h per experiment × 180 experiments) to one FCN model. Secondly, it is also interesting to compare the OVPC based approaches with the multi-view based approaches and definitely, time consumption would be dramatically increased. Fortunately, results of several multi-view based approaches [50,51,52,53,54] are available for comparison as shown in Table 4. In addition, this study focuses on one database and the five cities show unique characteristics in urban environment among western cities (America and Austria), while large databases with global cities [61,79] would be more general.

## 6. Conclusions

This paper proposes the concept of “one view per city” and conducts a proof-of-concept study on the segmentation of buildings in remote-sensing images. Five deep networks are evaluated on images of five cities in the Inria Aerial Image Labeling database. Experimental results suggest that the concept can be explored to decrease the number of images for model training and it enables us to achieve competitive performance in buildings segmentation with decreased time cost. In addition, several techniques to improve and to extend the concept for remote-sensing image analysis are suggested. The proposed concept can relieve the challenge of large-scale data annotation in deep learning based remote-sensing image segmentation. It might pave the way for multi-source remote-sensing image analysis.

## Figures and Tables

**Figure 1 sensors-20-00141-f001:**
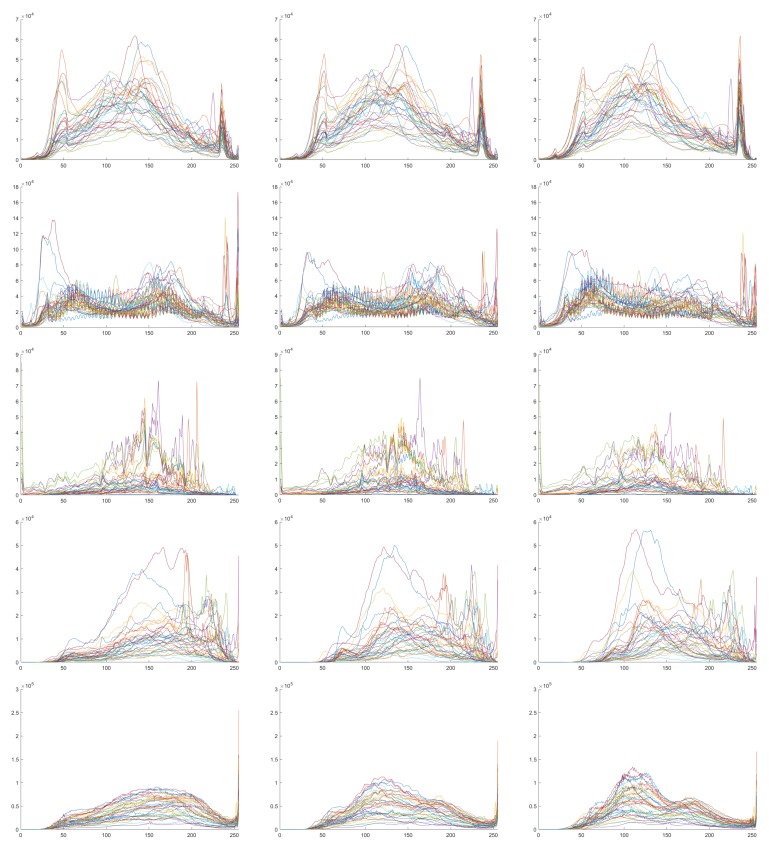
The intensity distribution of each city from Austin, Chicago, Kitsap County, Western Tyrol to Vienna (plots in each row) and each channel from R, G to B (plots in each column). In each plot, the horizontal axis shows the intensity range ([1, 255]) and the vertical axis shows the number of pixels to each intensity value.

**Figure 2 sensors-20-00141-f002:**
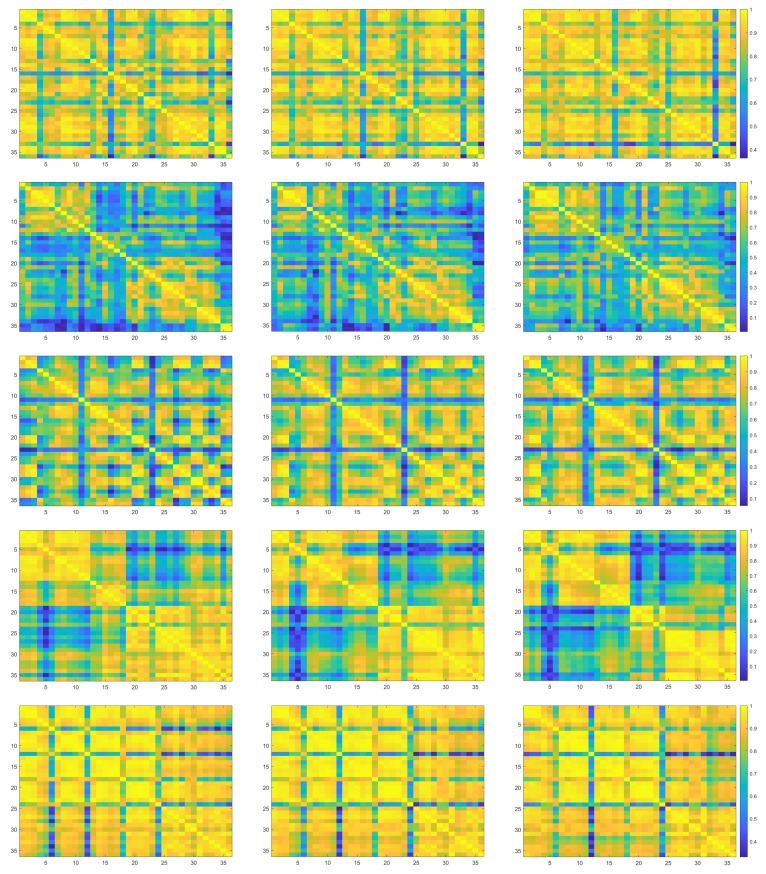
The pair-wise linear correlation coefficient matrix of each city from Austin, Chicago, Kitsap County, Western Tyrol to Vienna (plots in each row) and each channel from R, G to B (plots in each column). In each plot, both the horizontal and the vertical axis shows the RS image index.

**Figure 3 sensors-20-00141-f003:**
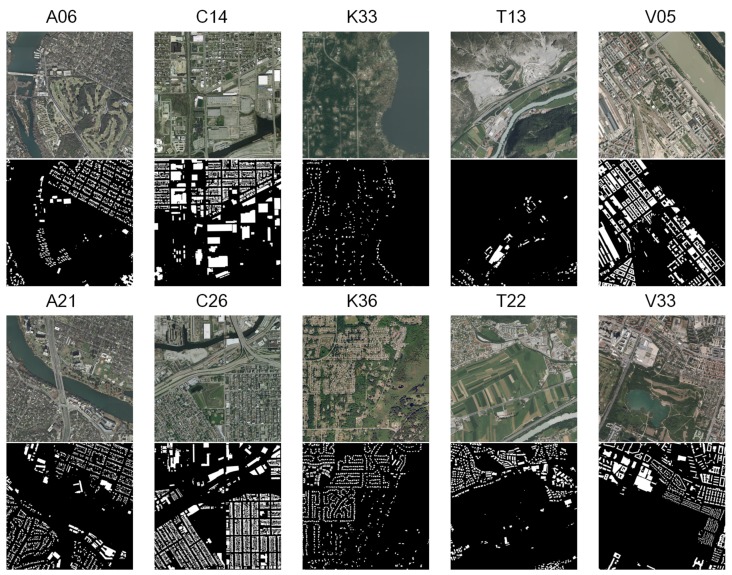
Visual comparison of remote-sensing images from different cities (Austin, Chicago, Kitsap County, Western Tyrol and Vienna). The binary images under each remote-sensing images correspond to the annotated labels of buildings regions.

**Figure 4 sensors-20-00141-f004:**
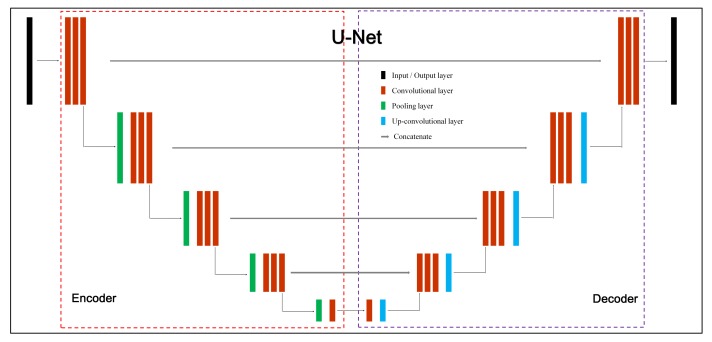
The architecture of U-Net. It consists of an encoder and a decoder network symmetrically. The main blocks of the network include convolutional layers, pooling layers, up-convolutional layers and concatenate parts. The concatenate parts aim to propagate original information for accurate image restoration.

**Figure 5 sensors-20-00141-f005:**
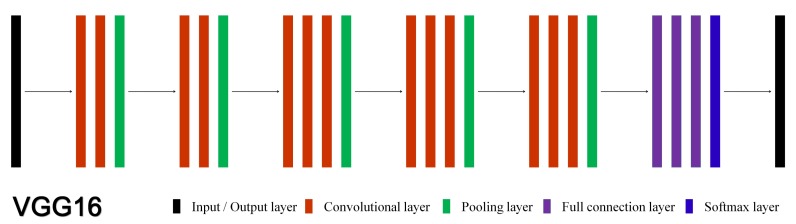
The 16-layer VGG net. Different colors stand for different layers. Besides the input and the output layer, the net consists of 13 convolutional layers, 5 pooling layers, 3 full-connection layers and 1 softmax layer.

**Figure 6 sensors-20-00141-f006:**
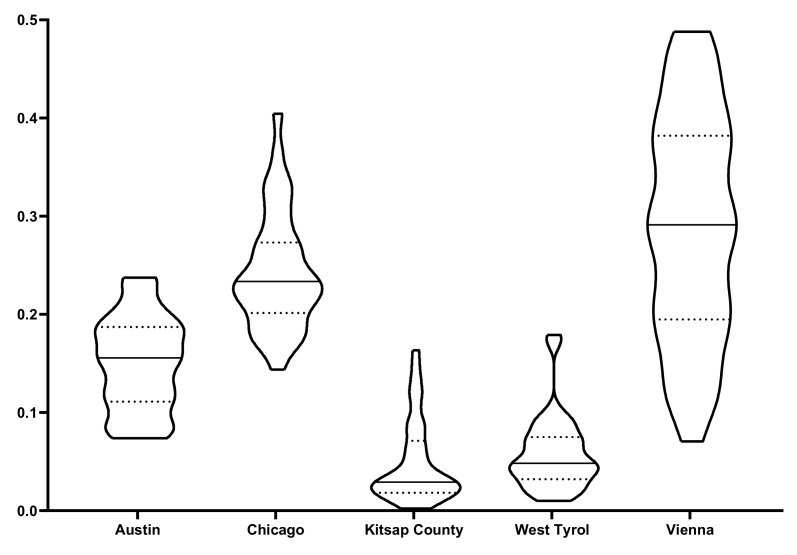
Distribution of brr values. The brr value reflects the coverage of buildings region in each remote-sensing image. The x-axis indicates cities and the y-axis shows brr values. In violin plots, solid lines denote median values and dashed lines stand for quartiles. It indicates that the city Vienna has the highest buildings density, followed by Chicago and Austin, while remote-sensing images of Western Tyrol and Kitsap County are with buildings thinly scattered.

**Figure 7 sensors-20-00141-f007:**
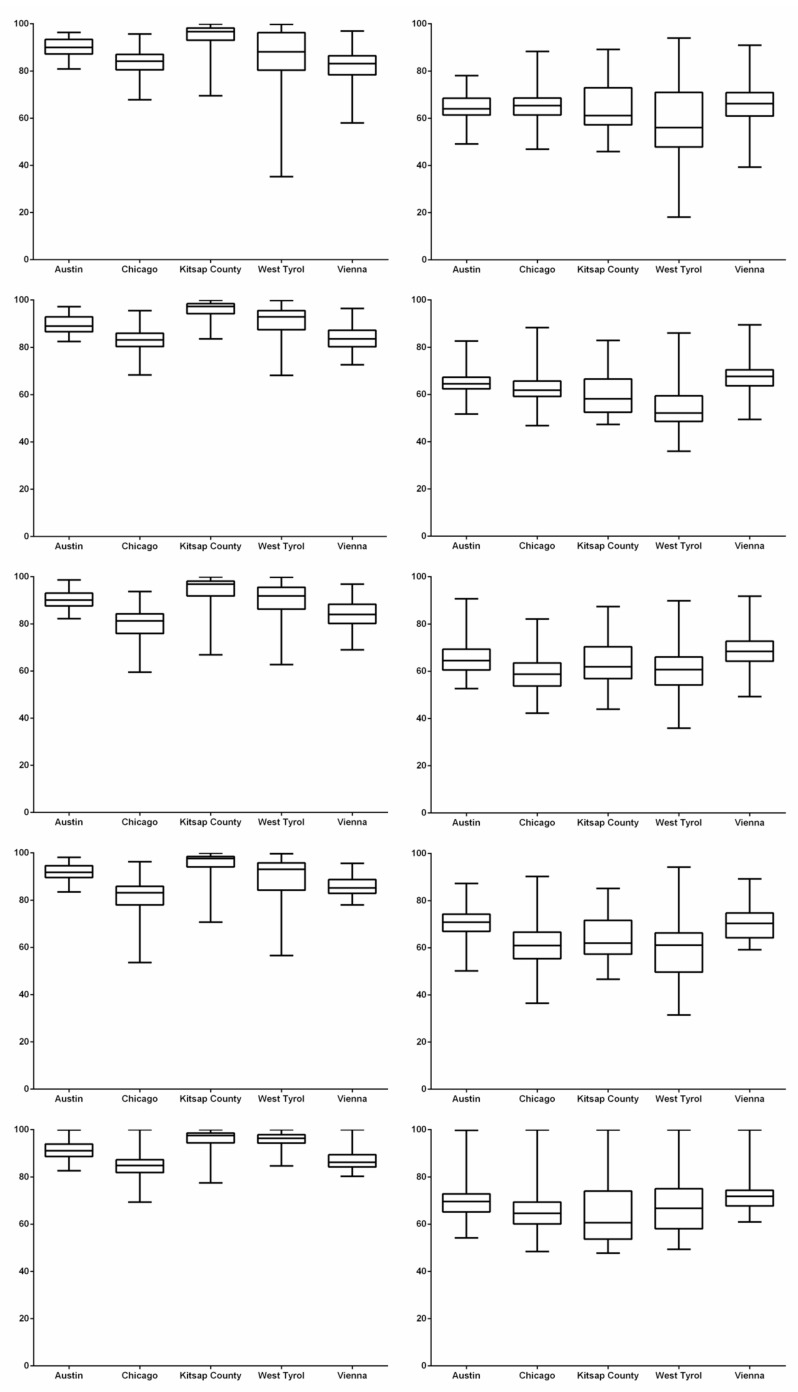
Comparison of fully convolutional neural network (FCN) models on OVPC based buildings segmentation using RS images (intra-city test). Each row indicates results from a FCN model from FCN8, FCN32, SegNet, TernausNet to U-Net, and each column shows a performance metric from ACC to intersection over union (IoU). In each box and whisker plot, the x-axis denotes the cities and the y-axis stands for metric values (%).

**Table 1 sensors-20-00141-t001:** One view per city (OVPC) based experiment design for buildings segmentation in RS images.

Training	Image No.	Intra-City Test	Image No.	Inter-City Test	Image No.
Austin	1	Austin	35	Other cities	36 × 4
Chicago	1	Chicago	35	Other cities	36 × 4
Kitsap County	1	Kitsap County	35	Other cities	36 × 4
West Tyrol	1	West Tyrol	35	Other cities	36 × 4
Vienna	1	Vienna	35	Other cities	36 × 4

**Table 2 sensors-20-00141-t002:** Segmentation accuracy (ACC) values of OVPC based buildings segmentation in RS images (mean ± std, %). The highest mean values in each intra-city test are in bold.

	Austin	Chicago	Kitsap County	West Tyrol	Vienna
FC8	89.89 ± 3.87	83.20 ± 5.77	94.56 ± 6.06	86.17 ± 12.12	82.45 ± 6.77
FC32	89.45 ± 3.72	82.83 ± 4.76	**95.91 ± 3.84**	90.43 ± 7.63	83.78 ± 5.47
SegNet	90.08 ± 3.66	79.41 ± 7.18	94.04 ± 6.68	90.25 ± 6.85	84.07 ± 5.78
TernausNet	**91.54 ± 3.57**	81.16 ± 7.78	95.17 ± 5.75	88.67 ± 10.45	85.91 ± 4.24
Unet	91.07 ± 3.78	**84.41 ± 5.41**	95.62 ± 4.70	**95.70 ± 3.11**	**87.05 ± 4.20**

**Table 3 sensors-20-00141-t003:** IoU values of OVPC based buildings segmentation in RS images (mean ± std, %). The highest mean values in each intra-city test are in bold.

	Austin	Chicago	Kitsap County	West Tyrol	Vienna
FC8	64.53 ± 5.69	64.86 ± 7.45	**64.07 ± 9.27**	58.69 ± 15.53	65.67 ± 8.13
FC32	64.86 ± 5.01	62.19 ± 5.83	59.47 ± 7.96	54.22 ± 9.75	66.90 ± 6.92
SegNet	64.96 ± 6.24	58.47 ± 7.29	63.14 ± 8.68	61.41 ± 10.68	68.34 ± 7.73
TernausNet	**70.08 ± 6.47**	60.47 ± 8.94	63.69 ± 8.34	58.55 ± 11.86	69.97 ± 6.30
Unet	69.60 ± 7.01	**65.36 ± 8.58**	63.31 ± 11.66	**67.63 ± 10.99**	**72.11 ± 6.43**

**Table 4 sensors-20-00141-t004:** Recent outcomes on the Inria Aerial Image Labeling (IAIL) database based on intra-city test (mean value, %).

		Austin	Chicago	Kitsap County	West Tyrol	Vienna
MLP in [40] (baseline)	ACC	94.20	90.43	98.92	96.66	91.87
	IoU	61.20	61.30	51.50	57.95	72.13
FCN in [40]	ACC	92.22	88.59	98.58	95.83	88.72
	IoU	47.66	53.62	33.70	46.86	60.60
Mask R-CNN in [50]	ACC	94.09	85.56	97.32	98.14	87.40
	IoU	65.63	48.07	54.38	70.84	64.40
Two-level U-Net in [51]	ACC	96.69	92.40	99.25	98.11	93.79
	IoU	77.29	68.52	72.84	75.38	78.72
Multi-task SegNet in [52]	ACC	93.21	**99.25**	97.84	91.71	**96.61**
	IoU	76.76	67.06	**73.30**	66.91	76.68
MSMT in [53]	ACC	95.99	92.02	99.24	97.78	92.49
	IoU	75.39	67.93	66.35	74.07	77.12
GAN-SCA in [54]	ACC	**97.26**	93.32	**99.30**	**98.32**	94.84
	IoU	**81.01**	**71.73**	68.54	**78.62**	**81.62**
Unet ^a^	ACC	94.34	88.72	98.67	97.52	92.48
	IoU	72.48	68.14	67.50	72.16	74.35
Unet (OVPC)	ACC	91.07	84.41	95.62	95.70	87.05
	IoU	69.60	65.36	63.31	67.63	72.11

^a^ The Unet is with the same network architecture and parameters as the OVPC based Unet, while it is tested with the first 5 images of each city and trained with the other 31 images.

**Table 5 sensors-20-00141-t005:** ACC values of OVPC based Unet on buildings segmentation in RS images (mean ± std, %). The inter-city test results are in bold.

	Austin	Chicago	Kitsap County	West Tryol	Vienna
Austin	**91.07 ± 3.78**	88.72 ± 6.72	82.47 ± 8.58	84.13 ± 7.32	90.04 ± 4.33
Chicago	82.68 ± 6.82	**84.41 ± 5.41**	78.69 ± 8.36	80.25 ± 6.53	82.65 ± 4.75
Kitsap County	92.52 ± 6.43	91.75 ± 7.32	**95.62 ± 4.70**	93.48 ± 3.79	89.82 ± 8.09
West Tryol	93.21 ± 8.52	90.42 ± 7.12	94.85 ± 5.84	**95.70 ± 3.11**	91.46 ± 8.42
Vienna	82.1 ± 5.47	82.78 ± 7.38	78.49 ± 6.12	80.54 ± 6.79	**85.91 ± 4.24**

**Table 6 sensors-20-00141-t006:** IoU values of OVPC based Unet on buildings segmentation in RS images (mean ± std, %). The inter-city test results are in bold.

	Austin	Chicago	Kitsap County	West Tryol	Vienna
Austin	**69.60 ± 7.01**	62.58 ± 10.23	58.62 ± 12.35	60.27 ± 14.23	54.73 ± 15.29
Chicago	60.18 ± 10.32	**65.36 ± 8.58**	57.54 ± 14.46	59.73 ± 10.89	63.08 ± 13.65
Kitsap County	59.73 ± 10.54	56.26 ± 16.46	**63.31 ± 11.66**	61.35 ± 12.36	52.83 ± 15.23
West Tryol	61.42 ± 12.49	58.76 ± 13.72	64.45 ± 14.36	**67.63 ± 10.99**	52.10 ± 16.47
Vienna	64.48 ± 8.25	66.29 ± 9.21	55.23 ± 10.23	57.58 ± 8.82	**72.11 ± 6.43**

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
