# Peer review of "One View Per City for Buildings Segmentation in Remote-Sensing Images via Fully Convolutional Networks: A Proof-of-Concept Study"

_sensors, 2019, doi:10.3390/s20010141_

Round 1

Reviewer 1 Report

This manuscript developed a "one view per city" concept to segment buildings from remote sensing images. Particularly, the IAILD, a dataset with five cities, including Austin, Chicago, Vienna, Western Tyrol, and Kitsap County, was applied to prove the concept. This is a well-written manuscript, and the concept seems to be reasonable. The academic contribution, however, is modest, and the authors cannot prove the "one view per city" concept. My major concerns are as follows.

1) The selection of the IAILD: I understand that one of the reasons is the groundtruthing. These five cities or regions, however, are similar to each other in some sense. In order to prove the concept, the authors need to choose a large number of cities globally, not just five western cities with similar building structures.

2) The necessity of such analyses: The advantages of deep learning are associated with large labelled training data, and the authors attempted to mitigate the advantages with consideration of computation cost. To prove the effectiveness of the developed method, they have to compare to the traditional deep learning methods, and analyze their pros and cons.

In general, this is an interesting idea, but I have major concerns regarding the necessary of such an approach, and the application of the approach in five cities with similar building styles is questionable.

Author Response

The authors are very grateful for your valuable comments and suggestions on our manuscript. We have tried our best to revise our manuscript accordingly, hoping that the correction will meet with approval.

Reviewer 2 Report

This paper proposes new idea to increase efficiency of CNN-based buildings segmentation on remote sensing images. The idea is in using of single RS image with ground truth segmentation to train segmentation CNN for whole city, this approach is called One View per City, OVPC. Authors have validated the approach with different state-of-the-art CNN architectures.

The idea looks elegant, but I am concerned about the following issues, especially about justification of the proposed approach.

Major concerns.

First, OVPC approach must be compared with multi-view and a few-view per city segmentation. Authors show the accuracy (ACC) for OVPC, the ACC values are not bad, but it necessary to compare it with the ACC values for the same CNN trained over multi images of the city. It is necessary to show advantages and disadvantages of proposed OVPC approach, at least in accuracy, in training, and inference speed. It is also necessary to show the ACC values of CNN trained on one city and evaluated on another. E.g. what ACC will CNN trained on Tirol shows on Vienna. The intuition of OVPC concept provided in Section 2 shows the correlation between the same cites views, however this correlation is not always high, e.g. for Chicago. Plot captions need to have more details. Axes captions are sometimes omitted (e.g. Fig 1).

There are some concerns in experimental section.

Why authors used only two images per city for training, what about the standard cross validation procedure? The authors refer to “limited computational resources” (L157), however, the training time was not shown. They train the CNN only for 100 iterations (L166). I’m not sure about the huge computational complexity of this training procedure. I’m not sure, that the paper’s subject is belonging to the scope of the MDPI Sensors journal, I think the MDPI Remote Sensing is more appropriate one.

Some minor concerns.

L179 - Keras over python 2.7 – why do you use python 2.7 and which backend do you use with Keras?

Some phrases need to be rewritten e.g.

L4 - However, it requires massive images for hyper-parameter optimization. L116 - can be made most of to address L120 - database IAILD

Thus, I think that the paper could not be published in the MDPI Sensors in its current state.

Sincerely, The Reviewer.

Author Response

(The authors gave the same response as above.)

Reviewer 3 Report

This manuscript presents a really interesting paper and it is well presented. With aim to improve the quality of this manuscript, these are my suggestions:

The authors argue “Quantification of these changes can be used to balance economic and environmental benefits and most importantly, it can be used to support the sustainable urban development.” I am totally agree. In fact, I recommend to extend a little bit this part in the introduction. I suggest you review some studies that relate urban metrics with urban typologies and land use classification. Why LiDAR data are not estimated within this study? They are really valuable for changes detection, especially in urban environments. They are used for land use mapping and control of changes in some studies. In my eyes, this should be at least cited. For knowing more about LiDAR potentialities in urban environments you could check studies related to “Airborne LiDAR point density analysis” I am totally agree with this affirmation “It gave clues on the relationship among urbanization, land use efficiency of urban expansion and population growth. Moreover, these factors are highly related to carbon emissions, climate change and urban environmental development that facilitate urban planning and management [5].” However, you should add some more studies related with this perspective. I think that should be convenient in the introduction part that the authors introduce the topic related to “changes detection” in urban environments. A fast review show how some of these studies are based on diverse techniques such as aerial imaginery (Automated spatiotemporal change detectionin digital aerial imagery), remote sensing (Change Detection in Urban Areas using Satellite Data), LiDAR (Detection of illegal constructions in urban cities) and even cartography (Changes in the urban model of the city of Valencia (Spain): an analysis from the point of view of the published cartography). Please, add this paragraph and include some references.

Author Response

(The authors gave the same response as above.)

Round 2

Reviewer 1 Report

I appreciate the authors' efforts to make the paper stronger. I am now happy to see it being accepted.

Author Response

Thanks for your approval.

Reviewer 2 Report

The paper was rewritten significantly. The authors have addressed most of my previous concerns. However, my main concern about the significance of the proposed contribution is still valid.

- Authors add table 4, with the accuracy of related works, but it is hard to compare Table 4 and Table 2 with their results. The networks in these tables are mostly different. Similar networks, U-Net and FCN trained using proposed method shows worse results. Overall accuracy in Table 2 is worse than the baseline of Table 4.

- So, the main question of what is benefice of using OVPC, how much is training speedup comparing with the traditional approach, how much is accuracy trade-off? Speedup and accuracy must be compared for the same network architecture. It must be compared at least on couple on network architectures, e.g. MLP and U-Net.

- I repeat my concern about cross-validation. “Why authors used only two images, what about the standard cross-validation procedure?”

- I repeat my concern about validation on different cities, this validation will show, is the OVPC concept really work or not.

“It is also necessary to show the ACC values of CNN trained on one city and evaluated on another. E.g. what ACC will CNN trained on Tirol shows on Vienna.”

Thus, my decision is still weak reject.

Sincerely, The Reviewer

Author Response

The authors are very grateful for your valuable comments and suggestions on our manuscript. We have tried our best to revise our manuscript accordingly, hoping that the correction will meet with approval. Four files are submitted, one is labeled “Point-to-point response” (this file), one is the manuscript “ovpc”, one is a supplementary file labeled “ovpc_mark”, and the compressed file of the manuscript “SensorsV3”. In addition, the point-to-point response to each reviewer has also been uploaded.

The point-by-point responses to all comments of the reviewers are attached.
